# Modifications of Phytohormone Metabolism Aimed at Stimulation of Plant Growth, Improving Their Productivity and Tolerance to Abiotic and Biotic Stress Factors

**DOI:** 10.3390/plants11243430

**Published:** 2022-12-08

**Authors:** Beatrycze Nowicka

**Affiliations:** Department of Plant Physiology and Biochemistry, Faculty of Biochemistry, Biophysics and Biotechnology, Jagiellonian University, Gronostajowa 7, 30-387 Kraków, Poland; beatrycze.nowicka@uj.edu.pl

**Keywords:** phytohormones, transgenic plants, biotic stress, abiotic stress, growth regulators

## Abstract

Due to the growing human population, the increase in crop yield is an important challenge for modern agriculture. As abiotic and biotic stresses cause severe losses in agriculture, it is also crucial to obtain varieties that are more tolerant to these factors. In the past, traditional breeding methods were used to obtain new varieties displaying demanded traits. Nowadays, genetic engineering is another available tool. An important direction of the research on genetically modified plants concerns the modification of phytohormone metabolism. This review summarizes the state-of-the-art research concerning the modulation of phytohormone content aimed at the stimulation of plant growth and the improvement of stress tolerance. It aims to provide a useful basis for developing new strategies for crop yield improvement by genetic engineering of phytohormone metabolism.

## 1. Introduction

Due to the constantly growing human population, ensuring high crop productivity is an important challenge for 21st century agriculture. The research aimed at obtaining high-yielding varieties is being carried out [1]. Another important issue is to obtain varieties displaying enhanced tolerance to biotic and abiotic stresses that cause significant loss of yield. Among the abiotic stresses, the most important are drought, thermal stress (too high or too low temperature), light stress, salt stress, and stress caused by environmental pollution, e.g., by heavy metal ions. Due to anthropogenic climate change, an increase in abiotic-stress-evoked losses of crop yield is expected in the near future [2]. Apart from abiotic factors, the biotic ones, such as pathogens, competing plants, parasites, and herbivores, also limit plant growth and productivity.

The application of mineral fertilizers, herbicides, and pesticides, as well as growing high-yielding varieties obtained via traditional breeding methods, enabled a significant increase in crop productivity during the second half of the 20th century [3]. For example, the average cereal yield in 1951 was 1.2 t/ha, while in 1993 it was 2.3 t/ha [4]. However, this yield increase has slowed down in the 21st century. It is currently believed that for the most important crop species, further increases in their productivity obtained by traditional breeding methods are possible only to a small extent. For this reason, research based on genetic engineering became crucial for the future of agriculture [1]. In addition to experiments on transgenic organisms, extensive genome analyses of major crop species are also being carried out. Their goal is to identify quantitative trait loci (QTLs), which are genes determining quantitative traits [5].

Research on transgenic plants conducted over the past decades resulted in the development of various strategies of genetic modification aimed at obtaining lines with increased yield or improved tolerance to stress [6,7]. One of the promising research directions is associated with the modulation of phytohormone levels [2]. Phytohormones participate in the regulation of plant growth and development. They also play a role in response to environmental factors. These compounds include auxins, cytokinins, gibberellins, abscisic acid (ABA), ethylene, jasmonic acid (JA) and its derivatives, and brassinosteroids. Auxins, cytokinins, gibberellins, and brassinosteroids are considered particularly important for the regulation of plant growth and development, while JA, ABA, and ethylene play crucial roles in stress response. However, one needs to remember that growth-stimulating hormones participate in stress responses, while those primarily associated with the stress response are also involved in the regulation of various plant developmental processes, such as dormancy, fruit maturation, or senescence [8]. Plant hormones have pleiotropic effects. Furthermore, the result of their action often depends on cross-talk between various phytohormones and signaling molecules [9]. Phytohormones occur in plants at very low concentrations; their biosynthesis and degradation are strictly regulated. In some cases, reversible inactivation by conjugate formation is also possible [10].

This review presents the current state of research on the modulation of phytohormone content aimed at obtaining transgenic plants with traits favorable for the breeder, such as increased yield and improved tolerance to abiotic and biotic stress factors.

## 2. Strategies Applied in Phytohormone-Targeted Genetic Engineering

The research aimed at improving crop performance by modification of phytohormone metabolism and signaling starts with identification of the crucial genes. This is possible mainly due to the studies carried out on mutants or by comparing crop varieties displaying desirable traits with the other ones [10]. Gene and genome sequencing enables the identification of loci crucial for the observed effects. Analyses of phenotypes and detailed analyses at the biochemical level, i.e., determination of phytohormone content, enable scientists to discover gene functions. When the sequence and function of its product are known, bioinformatics provides tools to find homologues in other species. At this point, the plant transformation can be carried out to increase or decrease the level of a certain phytohormone. The increase in hormone level can be achieved by the overexpression of the gene encoding enzyme participating in the phytohormone biosynthetic pathway or silencing of the gene whose product catalyzes hormone degradation. The decrease can be achieved by silencing of the gene crucial for phytohormone biosynthesis or by overexpression of the gene whose product is involved in hormone degradation. The manipulation of the genes encoding enzymes carrying out phytohormone conjugation was also carried out [10]. Sometimes, the increase in phytohormone level may be achieved by the enhanced production of an enzyme catalyzing the formation of a metabolite that serves as a phytohormone precursor (for example, xanthophyll precursors of ABA biosynthesis) or a cofactor needed by the hormone-synthetizing enzyme (for example, molybdenum cofactor required for abscisic aldehyde oxidase activity) (see subchapter concerning ABA). The research on the engineering of phytohormone transport was also carried out (see subchapter about auxins). The significant progress in our understanding of phytohormone signaling opens a wide range of possibilities, as various elements of signaling cascades, transcription factors, and miRNAs are emerging targets for potential modifications. These strategies have been mentioned, but their detailed description is beyond the scope of the present review. In the early research, scientists used strong, constitutive promoters to provide the overexpression of desired genes. The discovery of tissue-specific, developmental-stage-specific, and stress-responsive promoters enabled the improved control of the time and site of transgene expression [10]. Furthermore, artificial promoters have been developed. Considering gene silencing, various constructs may be applied, including antisense sequences, 3′-untranslated regions, and hairpin constructs. The recent development of the CRISPR/Cas9 system paved the way for extensive genome editing (see subchapter about future perspectives).

## 3. Auxins

Auxins play a key role in the regulation of plant growth and development, therefore, modulation of their biosynthesis and signaling has been a subject of intensive research [11]. The most important auxin is indole-3-acetic acid (IAA). The substrate for its synthesis is tryptophan, but this amino acid can be transformed in various ways, all of them leading to the same final product. One of the known pathways of IAA synthesis involves two steps: tryptophan oxidation to indole-3-pyruvate, followed by oxidation of indole-3-pyruvate to IAA. The second reaction is catalyzed by monooxygenase encoded by *YUCCA* genes. Alternatively, indole-3-pyruvate can be converted to indole-3-acetaldehyde and then to IAA. Another known IAA biosynthetic pathway leads through indole-3-acetamide. There is also a pathway specific to the Brassicaceae family, for which the intermediate is indole-3-acetonitrile [12]. The concentration of auxins depends on the synthesis, degradation, and transport of this phytohormone; it is also regulated by conjugation. In the latter case, the important role is fulfilled by enzymes encoded by the *GH3* family of genes [13].

The results of the experiments carried out on transgenic lines with changed auxin content are summarized in Table 1. In terms of practical application, the modification of auxin levels in developing flowers seems to be the most promising direction. This effect is achieved by the expression of bacterial gene encoding tryptophan monooxygenase (e.g., *iaaM* from *Pseudomonas syringae pv. savastanoi*) under the control of an ovule-specific promoter. Increased auxin content stimulates the growth of generative shoots and fruits of transgenic plants. The application of tissue-specific promoters is better than the use of constitutive ones because the enhanced auxin synthesis occurring in whole plants often leads to undesirable developmental disorders (Table 1). Apart from the genes directly involved in auxin biosynthesis and degradation, the genes whose products play regulatory roles can also be targets of manipulation. It was observed that the expression of *OsIAA6* was highly induced by drought stress. Transgenic rice with overexpression of this gene under the control of a constitutive promoter displayed enhanced expression of *YUCCA* genes and was more tolerant to drought [14].

The research carried out on mutants is also important for better understanding of auxin roles. *Arabidopsis thaliana* mutant *yuc7-1D*, with the altered gene *YUCCA7*, displayed a phenotype characteristic for plants with auxin overproduction: tall stems and curled, narrow leaves. It was also more tolerant to drought when compared to control plants [15]. *A. thaliana* mutant *arf2* producing inactive Auxin Response Factor protein (ARF) developed longer and thicker flower shoots, larger and darker leaves, and larger seeds [16]. The *brachytic2* (*br2*) maize mutant with impaired auxin transport in the stem had shorter internodes. This observation may be of practical significance because dwarfism is a desirable trait of cereals as it provides a more favorable ratio of grain biomass to shoot biomass and increases lodging resistance [17]. The modification of soybean *GmPIN1* using CRISPR/Cas9 method resulted in plants displaying changed architecture [18].

**Table 1 plants-11-03430-t001:** Summary of the results of the experiments on transgenic plants with changed auxin concentration or transport. MDA, malonyldialdehyde; RWC, relative water content.

Protein	Promoter and Gene	Species	Phenotype of Transgenic Plants Compared to Control Lines	Reference
indole-3-pyruvate monooxygenase	*35S::AtYUCCA6*	potato (*Solanum* *tuberosum*)	- changed morphology: increased height, erect leaves, narrow downward-curled leaves - increased longevity - increased drought tolerance: far less pronounced wilting symptoms, increased water content in the leaves of stress-exposed plants (plants not watered for 18 days), ability to recover after rewatering while control plants were dying - decreased tuber biomass per plant	[19]
	*SWPA2::AtYUCCA6* oxidative stress-induced promoter	poplar (*Populus alba × P. glandulosa*)	- rapid shoot growth, retarded main root development, increased root hair formation - disturbed leaf morphology: folded leaves, elongated petioles, long internodes - delayed hormone- and dark-induced senescence of detached leaves - increased drought tolerance: less pronounced symptoms of wilting (plants not watered for 6 days)	[20]
	*SWPA2::AtYUCCA6*	sweet potato (*Ipomea batatas*)	- changed morphology: narrow, downward-curled leaves, increased height, increased node number - increased drought tolerance: higher RWC and lower content of MDA in stressed plants (plants not watered for 16 days) - lower storage root formation	[21]
tryptophan monooxygenase from bacteria	*DefH9::iaaM* ovule-specific promoter	tomato (*Lycopersicon esculentum*)	- parthenocarpic fruit development	[22]
	*DefH9::iaaM*	raspberry (*Rubus idaeus*) strawberry (*Fragaria × ananassa*)	- more inflorescences, flowers and fruits - bigger fruits - increased fruit biomass normalized per plant - parthenocarpic fruit development	[23]
	*DefH9::iaaM*	grape vine (*Vitis vinifera*)	- more inflorescences per shoot - increased berry number per bunch	[24]
	*FBP7::iaaM* flower-specific promoter	cotton (*Gossypium hirsutum*)	- enhanced fiber yield (field trials) - increased fiber quality	[25]
	*B33::tms1* tuber-specific promoter	potato (*Solanum tuberosum*)	- enhanced tuberization	[26]
enzymes catalyzing auxin conjugation	*35S::OsGH3.1*	rice (*Oryza sativa*)	- dwarfism - increased tolerance to fungal infection (less pronounced symptoms of infection by *Magnaporthe grisea*)	[13]
	*Ubi1::OsGH3-2*	rice (*Oryza sativa*)	- changed morphology: dwarfism, smaller leaves, fewer crown roots and root hairs, smaller panicles - decreased tolerance to drought: earlier and more pronounced wilting symptoms, much lower survival rate (seedlings not watered for 4 days) - increased tolerance to cold stress: less pronounced symptoms on leaves, significantly increased survival rate after recovery (5 days in 4 °C)	[27]
auxin transporter	*35S::ZmPIN1a*	maize (*Zea mays*)	- increased number of lateral roots and dry weight of roots - lower shoots, shorter internodes - increased seed biomass per plant under high-density cultivation - increased drought tolerance: wilting symptoms of stress-exposed plants appeared later, the majority of transgenic plants survived drought, while most of the wild type plants died (seedlings not watered for 5 days) - increased seed biomass per plant under drought conditions (field trials)	[28]

The knowledge concerning auxin-initiated signaling pathways has intensively broadened during the last decades. The modification of these pathways allows us to obtain plants with desired traits. For example, overexpression of the gene encoding auxin-induced protein ARGOS in *A. thaliana* resulted in the stimulation of cell proliferation and an increase in organ size [29]. Overexpression of maize ARGOS1 (ZAR1) stimulated organ growth, enhanced grain yield, and drought stress tolerance in transgenic maize [30]. Overexpression of Auxin Response Factor 19 (ARF19) homolog from *Jatropha curcas* in *A. thaliana* and *J. curcas* increased seed size and yield [31]. Expression of gene *IbARF5* from sweet potato under the *35S* promoter in *A. thaliana* resulted in enhanced tolerance to drought and salinity in transgenic plants [32]. On the other hand, downregulation of Auxin Response Factor 4 (ARF4) in tomato increased tolerance to salinity and drought stress [33]. Similar results were obtained by Chen et al. [34]. Overexpression of the gene *OsAFB6*, encoding an auxin receptor, in rice resulted in increased grain yield per plant both in short day and long day conditions [35]. Another auxin receptor, *AFB3*, when overexpressed in Arabidopsis caused the increase in salt stress tolerance [36]. Transgenic maize overexpressing Auxin Binding Protein 1 (ABP1) was more resistant to sugarcane mosaic virus [37].

## 4. Cytokinins

Cytokinins are another class of phytohormones necessary for plant growth stimulation and controlling many developmental processes. They also participate in the regulation of plant senescence. Prolonging organ longevity due to cytokinin action enables longer biomass production by the plant [38]. Among the enzymes participating in cytokinin biosynthesis, the main targets of genetic engineering are as follows: isopentenyl transferase (IPT) catalyzing condensation of isoprenoid residue with adenine nucleotide, cytokinin dehydrogenase (CKX) involved in the degradation of these phytohormones, and glycosyl transferases converting cytokinins into their conjugates [38,39,40,41]. It was observed that mutations in *ckx* genes led to the increase in cytokinin content. Arabidopsis *ckx3 ckx5* double mutant formed larger inflorescences, floral meristems, and flowers and displayed increased seed yield per plant [42]. Similarly, *ckx3 ckx5* mutants of oilseed rape showed an increased cytokinin concentration that resulted in larger and more active inflorescence meristems, increased amounts of flowers and ovules and slightly increased seed yield [43]. Natural variations in soybean *GmCKX7-1* were linked to altered cytokinin profiles and yield characteristics [44].

The first attempts of genetic engineering of cytokinin metabolism were carried out in the 1990s. Gan and Amasino [45] transformed tobacco with the *IPT* gene under the control of *SAG12* promoter from *A. thaliana*, responsible for triggering gene expression in senescing leaves. The obtained transgenic lines displayed a greater number of flowers and seeds, delayed leaf senescence, and enhanced biomass production [45]. Due to the success of this strategy, promoters from the SAG family have been often used for plant genetic engineering. However, it has to be mentioned that in some cases, the delayed senescence of older leaves delayed nutrient allocation to seeds and storage organs. As a result, no yield increase and sometimes even yield reduction were observed. Furthermore, in the situation of nitrogen shortage, it was observed that old non-senescing leaves started to compete with younger leaves, which disturbed nitrogen recycling in plants. In rice, early senescing cultivars have a higher yield than those which undergo senescence later [46]. Positive effects were obtained by crossing the transgenic line of *A. thaliana* overexpressing *CKX3* under the control of root-specific *PYK10* promoter and displaying enhanced root growth with the lines displaying enhanced leaf growth [47].

Another promoter involved in the regulation of gene expression during senescence but also during stress response is senescence associated receptor protein kinase promoter *(SARK*). Expression of the *IPT* gene under the control of this promoter allowed for obtaining transgenic rice and tobacco with increased drought tolerance [48,49]. The examples of the experiments concerning the modulation of cytokinin metabolism are shown in Table 2.

The observed effects vary depending on the species and method used; not all of them are beneficial [50]. The application of inducible promoters responding to specific conditions allows better control of cytokinin content in transgenic plants. This allows us to avoid adverse effects occurring when too many of these phytohormones are synthesized in the plant [9]. Interestingly, the overexpression of *AGO2*, encoding protein belonging to the ARGONAUTE family and playing a role in the regulation of gene expression, led to the enhanced expression of cytokinin transporter BG3 and changed the pattern of cytokinin distribution in transgenic rice. This, in turn, resulted in an increase in grain length and salt tolerance [51]. Interesting results were obtained by Wang et al. [52], who used CRISPR/Cas gene editing to introduce changes into the cytokinin biosynthetic gene *OsLOG5*. The researchers managed to obtain rice lines with improved yield properties under drought stress when compared to stressed control [52].

It is noteworthy that in many cases researchers managed to obtain transgenic lines with increased biomass production or seed yield, but also more tolerant to abiotic stresses, such as drought and salinity (Table 2). However, it needs to be emphasized that there are some inconsistencies between the literature data, because the increased tolerance to some abiotic stresses was reported for plants with both increased and decreased cytokinin content. It needs to be remembered that in the experiments on transgenic plants, various species and promoters were used; there were also differences in the stress conditions applied [53].

Furthermore, cytokinins are a group of compounds, including *trans*-zeatin; *cis*-zeatin; N^6^-isopentenyladenine; dihydrozeatin; N^6^-benzylaminopurine; kinetin; *ortho*-, *meta*-, *para*-topolins; and ribosides of above-mentioned compounds [54]. It is already known that particular cytokinins vary in sites and timing of their production and degradation, transport routes, signaling pathways, and activity [54,55]. In *A. thaliana*, *trans*-zeatin and isopentenyladenine are the most active forms, present in higher concentrations than other cytokinins [54]. *Trans*-hydroxylated cytokinins, namely, the *trans*-zeatin-type, are synthesized in the roots and transported to the shoots in xylem sap. They are thought to play an important role as a nitrogen-supply signal in stimulation of the shoot growth. On the other hand, N^6^-isopentenyladenine and *cis*-zeatin-types are predominant in the phloem sap of *A. thaliana*. These species are thought to participate in systemic shoot-to-root signaling in cooperation with other signaling molecules [56]. The understanding of the specificity of certain cytokinin types synthesis, transport, and signaling is crucial for the successful genetic engineering of these phytohormones.

Plant responses aimed at restoring the homeostasis of cytokinin levels and signaling were also observed in plants with changed biosynthesis or degradation of these hormones. In some of the experiments, the cytokinin content in transgenic lines was not assessed, while in some others, the methods of cytokinin measurements were questioned by other scientists [55]. The intensive research on cytokinin synthesis, degradation, transport, and signaling is being carried out, which should enable us to explain these effects in the future.

**Table 2 plants-11-03430-t002:** Summary of the results of the experiments on transgenic plants with changed cytokinin concentration. APX, ascorbate peroxidase; CAT, catalase; Chl, chlorophyll; CKX, cytokinin dehydrogenase; IPT, isopentenyl transferase; O_2_^•−^, superoxide; POX, peroxidase; PSII, photosystem II; ROS, reactive oxygen species; RWC, relative water content; SOD, superoxide dismutase.

Enzyme	Promoter and Gene	Species	Phenotype of Transgenic Plants Compared to Control Lines	Reference
IPT	*Ubi::IPT*	tall fescue (*Festuca arundinacea*)	- increased tillering - increased tolerance to cold stress: delayed senescence in plants grown outdoors, decreased electrolyte leakage in detached leaves kept at temperature range 0 to –28 °C	[57]
	*35S::IPT*	wheat (*Triticum aestivum*)	- increased tolerance to flooding: less pronounced growth inhibition during flooding and higher yield after recovery (plants flooded for 14 days)	[58]
	*Wild type/35S::IPT* (scion/rootstock)	tomato (*Lycopersicon esculentum*)	- reduced root growth - increased tolerance to salt stress: increased fruit yield of stressed grafted plants (plants treated with 75 mM NaCl)	[59]
	*SAG12::IPT*	thale cress (*Arabidopsis thaliana*)	- increased tolerance to flooding: increased biomass and carbohydrate retention of waterlogged plants (plants flooded for 5 days) - improved recovery from waterlogging stress and after submergence stress (stress duration 5 days)	[60]
	*SAG12::IPT* *SAG13::IPT* senescence-specific promoters	tomato (*Lycopersicon esculentum*)	- suppression of leaf senescence - stem thickening, short internodal distances - loss of apical dominance - advanced flowering - slight increase in fruit weight per plant	[61]
	*SAG12::IPT*	cassava (*Manihot esculenta*)	- delayed leaf senescence observed both during dark induction of senescence of detached leaves in the greenhouse and during field trials - increased drought tolerance: reduced leaf senescence and wilting during water deficit (plants watered with lesser amount of water for 4 weeks)	[62]
	*SAG12::IPT* *HSP18::IPT* heat-stress-induced promoter	creeping bentgrass (*Agrostis stolonifera*)	- increased tolerance to heat stress: enhanced growth and root biomass of plants exposed to stress (plants grown at 35 °C/30 °C day/night for 10 days)	[63]
	*SAG12::IPT*	creeping bentgrass (*Agrostis stolonifera*)	- increased tolerance to drought (plants not watered for 21 days): more extensive root system, decreased MDA content and electrolyte leakage in roots, lower O_2_^•−^ and H_2_O_2_ levels in roots - increased antioxidant response in drought-exposed plants: increased ascorbate content, increased activity of SOD, CAT, APX, glutathione reductase, and dehydroascorbate reductase in roots	[64]
	*SAG12::IPT*	creeping bentgrass (*Agrostis stolonifera*)	- increased drought tolerance: suppression of drought-induced leaf senescence and root dieback, reduced wilting, lower MDA content, enhanced activity of SOD, CAT, POX (plants not watered for 2 weeks)	[65]
	*SAG12::IPT*	creeping bentgrass (*Agrostis stolonifera*)	- increased proline and soluble sugar content in drought-exposed plants (plants not watered till leaf RWC dropped to 47%)	[66]
	*SAG12::IPT*	eggplant (*Solanum melongena*)	- increased vegetative growth rate and fruit yield per plant - delayed leaf senescence - decreased MDA content, increased SOD and POX activity - increased drought tolerance: delayed chlorosis and wilting (plants were not watered) - increased cold tolerance: delayed chlorosis and wilting (plants kept in 4 °C)	[67]
	*SAG12::IPT* *DEG::IPT* dexamethasone-inducible promoter	thale cress (*Arabidopsis thaliana*)	- increased drought tolerance: faster and more vigorous recovery of stressed plants (plants not watered for 13 days)	[68]
	*GHCP::IPT* promoter belonging to SAG family	cotton (*Gossypium hirsutum*)	- delayed leaf senescence - increased lint yield, increased fiber quality (more uniform, stronger and longer fibers) - increased tolerance to salt stress: increased germination percentage under salt stress (on paper moistened with 250 mM NaCl) - increased dry biomass of plants exposed to salt stress (hydroponically grown seedlings exposed to 200 mM NaCl for 21 days)	[69]
	*SARK::IPT*	tobacco (*Nicotiana tabacum*)	- increased tolerance to drought: less severe stress symptoms, higher leaf water content and plant dry weight, improved recovery, increased seed yield of recovered plants (plants not watered for 2 weeks) - improved antioxidant defense in drought-exposed plants: increased ascorbate and glutathione content, decreased H_2_O_2_ level in leaves - only minimal seed yield loss of water-restricted plants (70% less watering)	[70]
	*SARK::IPT*	tobacco (*Nicotiana tabacum*)	- increased tolerance to water deficit: increased CO_2_ assimilation rate (70% reduced watering for 70 days)	[48]
	*SARK::IPT*	peanuts (*Arachis hypogaea*)	- increased drought tolerance: increased fresh and dry biomass of shoots and roots of stressed plants (plants not watered for 15 days, then watered with ¼ of the optimal water amount for 45 days) - increased shoot dry weight and seed weight per plant grown under water deficit (field trials)	[71]
	*SARK::IPT*	rice (*Oryza sativa*)	- increased drought tolerance: delayed wilting, increased total dry biomass and seed yield per plant in stress-exposed plants (plants not watered for 6–10 days before flowering phase or 2 weeks after flowering)	[49]
	*SARK::IPT*	rice (*Oryza sativa*)	- increased tolerance to drought: less pronounced stress symptoms, increased RWC and maximum quantum efficiency of PSII, no decrease in carbon and nitrogen assimilation and protein content (plants not watered for 3 days at pre anthesis) - increased sucrose and starch content in flag leaf, enhanced nitrate content, higher nitrate and nitrite reductase activity, and sustained ammonium content in drought-exposed plants	[72]
	*SARK::IPT*	cotton (*Gossypium* *hirsutum*)	- delayed leaf senescence (detached leaf assay) - increased drought tolerance: increased root and shoot biomass, Chl content, and photosynthetic rate under water deficit in the greenhouse; increased root and shoot biomass and cotton yield under water deficit in growth chamber (66% less watering)	[73]
	*SARK::IPT*	maize (*Zea mays*)	- increased drought tolerance: delayed wilting and leaf senescence, increased water content in stress-exposed plants, 30-fold higher average seed biomass per plant (plants not watered for 3 weeks)	[74]
	*SARK::IPT*	sweetpotato (*Ipomea batatas*)	- delayed senescence - improved tolerance to drought: improved growth characteristics and leaf RWC (plants exposed to various irrigation regimes for 96 days)	[75]
	*HMW::IPT* seed-specific promoter	tobacco (*Nicotiana tabacum*)	- increase in seed yield - increase in ethanol-insoluble carbohydrates and protein content	[76]
	*lectin::IPT* seed-specific promoter	tobacco (*Nicotiana tabacum*)	- increase in seed dry weight and protein content - faster growth of seedlings	[77]
	*TP12::IPT* flower-specific promoter	narrow-leafed lupin (*Lupinus* *angustifolius*)	- increased branching - increased total number of fruits (pods) in some lines	[78]
	*AtMYB32xs::IPT* developmental-process-related promoter	canola (*Brassica napus*)	- delayed leaf senescence both under controlled conditions and in the field - more flowers and siliques - increased yield (field trials)	[79]
	*AtMYB32xs-p::IPT*	wheat (*Triticum aestivum*)	- delayed leaf senescence - increased yield - improved drought tolerance: improved canopy green cover, lower canopy temperatures, higher leaf water potential (field trials)	[80]
	*rd29A::IPT* stress-induced promoter	tobacco (*Nicotiana tabacum*)	- increased tolerance to salt stress: delayed leaf senescence and decreased MDA content in stressed plants (plants exposed to 150 mM NaCl for 2 weeks)	[81]
	*AtCOR15a::IPT* cold-stress-induced promoter	sugarcane (*Saccharum officinarum* and *S. spontaneum* hybrids)	- increased tolerance to cold stress: less pronounced symptoms of leaf senescence (detached leaves exposed to 27 °C, 4 °C or 4 °C and then 0 °C), - increased Chl content, decreased MDA content, and electrolyte leakage in cold-stressed plants (plants were exposed to decreasing temperatures for acclimation, then incubated in 0 °C for 8 h and recovered for 24 h)	[82]
	*AtMT::IPT* stress-induced promoter	tobacco (*Nicotiana tabacum*)	- improved drought tolerance: less severe stress symptoms (plants not watered for 3 weeks) - improved tolerance to salt stress: less severe stress symptoms, faster recovery (plants watered with 100 mM NaCl for 10 days, then with 200 mM NaCl for 11 days)	[83]
	*PtRD26_pro_::IPT* promoter of senescence and drought-inducible transcription factor	poplar (*Populus* *tomentosa*)	- increased height, stimulated adventitious root generation - increased net CO_2_ assimilation - increased drought tolerance: less severe stress symptoms, higher levels of maximum quantum efficiency of PSII, RWC, net CO_2_ assimilation rate, stomatal conductance, and electron transfer rate, improved survival rate (plants not watered for 10 days)	[84]
*Trans*-zeatin synthetase	*hsp70::tzs* heat shock induced promoter	rapeseed (*Brassica napus*)	- reduced root system - increased height and branching - increased seed yield per plant	[85]
CKX	*35S::AtCKX1* *35S::AtCKX2* *35S::AtCKX3* *35S::AtCKX4*	tobacco (*Nicotiana tabacum*)	- retarded shoot development, dwarfed phenotype, small leaves - stimulated root growth	[86]
	*35S::AtCKX1* *35S::AtCKX2* *35S::AtCKX3* *35S::AtCKX4*	thale cress (*Arabidopsis thaliana*)	- reduced shoot growth - stimulated root growth	[87]
	*35S::CKX1* *35S::CKX2* *35S::CKX3* *35S::CKX4*	thale cress (*Arabidopsis thaliana*)	- reduced growth of some lines - increased tolerance to salt stress: less pronounced stress symptoms and improved survival rate (plants exposed to 200 mM NaCl for 6 days) - increased drought tolerance: less pronounced wilting symptoms and improved survival rate (plants not watered for 2 weeks)	[88]
	*35S::AtCKX3*	tomato (*Solanum* *lycopersicum*)	- smaller leaf area, decreased stomata density - decreased transpiration - increased drought tolerance: increased leaf water content (plants not watered for 4 days)	[89]
	*35S::MsCKX*	thale cress (*Arabidopsis thaliana*)	- enlarged root system - increased salt tolerance: improved root growth and seedling fresh weight (seedlings exposed to 100 or 150 mM NaCl for 7 days), improved survival rate and maximum quantum yield of PSII (plants watered with increasing concentrations of NaCl for 4 days, then with 350 mM NaCl for 10 days) - improved membrane properties and antioxidant defense under salt stress: decreased ion leakage, MDA content, H_2_O_2_ and O_2_^•−^ levels, increased proline content and SOD, CAT, POX activity (plants subjected to 150 mM NaCl for 10 days)	[90]
	*35S::AtCKX2*	rapeseed (*Brassica napus*)	- enlarged root system, longer primary roots, increased number of lateral and adventitious roots, increased root density, enhanced root-to-shoot ratio - no reduction in shoot growth - increased P, Ca, Mg, S, Zn, Cu, Mo, and Mn concentration in leaves - increased Chl content under Mg- and S-deficiency - improved phytoremediation capacity of Cd and Zn from contaminated medium and soil	[91]
	*35S::PpCKX1*	*Physcomitrella patens*	- larger size of protoplasts, curved protonemal tissues - delayed transition to gametophores, reduced number of spores - enhanced rhizoid development - improved tolerance to dehydration: increased survival rate after drying of protonemal tissues - improved tolerance to salt stress: improved growth (protonemal tissues exposed to 100 or 200 mM NaCl for 30 days)	[92]
	*Ubi::TaCKX1*	wheat (*Triticum aestivum*)	- increased spike number and grain number - lower 1000-grain weight	[93]
	*W6::CKX1* root-specific promoter	tobacco (*Nicotiana tabacum*)	- stimulation of root growth, increased ratio of root to shoot biomass - increased drought tolerance: increased survival rate (plants not watered for 26 days)	[94]
	*35S::CKX1* *WRKY6::CKX1* root-specific promoter	tobacco (*Nicotiana tabacum*)	- enlarged root system and dwarfism in line transformed with *35S* promoter construct - improved drought tolerance: higher water potential in lower leaves, more negative osmotic potential in leaves (plants not watered for 10 days) - lower leaf temperature in *35S:CKX1* line (plants exposed for 40 °C for 2 h)	[95]
	*RCc3::OsCKX4* root-specific promoter	rice (*Oryza sativa*)	- enhanced root development	[96]
	*RCc3::OsCKX4*	rice (*Oryza sativa*)	- increased Zn concentration in roots, shoots, and grains - increased grain yield per plot	[97]
	*bGLU::AtCKX1* root-specific promoter	barley (*Hordeum vulgare*)	- stimulated lateral root growth - improved drought tolerance: higher RWC, less pronounced decrease in yield (plants exposed to water deficit) - faster recovery and higher RWC in some drought-exposed transgenic lines (hydroponically grown plants were deprived of the growth medium for 24 h), improved growth after stress recovery (observed 2 weeks after severe stress application to hydroponically grown plants and 4 weeks after 3 days of watering withdrawal in soil-grown plants)	[98]
	*EPP::CKX1* *EPP::CKX2* root-specific promoter	barley (*Hordeum vulgare*)	- stimulation of root growth - increased concentration of various micro- and macro-elements in leaves - increased tolerance to drought: higher CO_2_ assimilation rate in stressed plants (plants were not watered until the soil moisture level dropped to 10%, this level was maintained for the next 2 weeks)	[99]
	*EPP::CKX1* *EPP::CKX2*	barley (*Hordeum vulgare*)	- increased Zn concentration in grains - increased Fe concentration in grains of some lines	[100]
	*RCc3::AtCKX1*	maize (*Zea mays*)	- stimulation of root growth - increased concentration of micro- and macro-elements in leaves: K, P, Mo, Zn	[101]
	*CaWRKY31::CaCKX6* root-specific promoter	thale cress (*Arabidopsis thaliana*) chickpea (*Cicer arietinum*)	- increase in lateral root number, root length, and root biomass in Arabidopsis and chickpea without any penalty to vegetative and reproductive growth of shoot - soil-grown chickpea exhibited higher root-to-shoot biomass - enhanced drought tolerance in soil-grown chickpea: increased shoot and root growth, increased CO_2_ assimilation rate (plants not watered for 40 days) - seed yield in some chickpea lines up to 25% higher with no penalty in protein content - higher levels of Zn, Fe, K, and Cu in transgenic chickpea seeds	[102]
	*RCc3:OsCKX5*	rice (*Oryza sativa*)	- stimulated root growth: greater volume, length, projection area, higher number of tips, enhanced surface area - no detrimental impact on shoot growth - increased root biomass, root to shoot ratio, deeper root system in plants grown on low-fertility soil - increased P, K, Ca, Mg, Zn, Fe concentration in roots - increased K, Mg, Fe, Zn concentration in shoots (not in all lines)	[103]
	*35S::MdCKX5.2*	thale cress (*Arabidopsis thaliana*)	- longer primary root, stimulation of lateral root development - increased tolerance to drought: less severe stress symptoms, significantly improved survival rate (plants not watered for 20 days) - increased tolerance to salt stress: less severe stress symptoms, longer primary roots and more lateral roots, increased fresh weight and Chl content (seedlings exposed to 100 or 150 mM of NaCl for 9 days)	[104]
	*OsCKX2* promoter::3′-UTR of *OsCKX2* target silencing	rice (*Oryza sativa*)	- increased grain number per plant	[105]
	*35S::HvCKX1* *35S::TaCKX1* hairpin target silencing	barley (*Hordeum vulgare*), wheat (*Triticum aestivum*) triticale	- increased seed yield, seed number per plant and 1000-grain weight in some lines - increased root biomass - lower shoots	[106]
	*35S::HvCKX2* hairpin target silencing	barley (*Hordeum vulgare*)	- increased height - increased spikes number - increased seed yield, seed number, and 1000-grain weight	[107]
	*35S::HvCKX1* hairpin target silencing	barley (*Hordeum vulgare*)	- increased root mass - increased seed yield	[108]
	*Ubi::shRNA-CX3* *Ubi::shRNA-CX5* hairpin target (*OsCKX2*) silencing	rice (*Oryza sativa*)	- delayed senescence - increased tillering - increased panicle number, grain yield per plant, and 1000-grain weight (field trials)	[109]
	*35S::GhCKX* hairpin target silencing	cotton (*Gossypium hirsutum*)	- delayed leaf senescence - more fruiting branches and bolls, increased seed size - increased seed yield and lint yield of moderately suppressed lines (per 25 m^2^ plot size)	[110]
	*35S::OsCKX2* antisense target silencing	rice (*Oryza sativa*)	- enhanced panicle branching, increased seed biomass per plant, increased 1000-grain weight - increased tolerance to salt stress: increased water content and higher shoots of stressed plants, at the end of stress-exposure wild type plants were dying (plants watered with 200 mM NaCl for 30 days) - less pronounced decrease in yield of plants exposed to salt stress at pre-flowering stage until maturity (plants watered with 100 mM NaCl)	[111]
	*Act1::HvCKX1* hairpin target silencing	wheat (*Triticum aestivum*)	- increased grain number per plant	[112]
	*Ubi1::HvCKX1* 5′ end of the ORF and 3′ UTR, target silencing *HvCKX1* knockout obtained by CRISPR/Cas9	barley (*Hordeum vulgare*)	- increased spike number and grain number per plant - decreased 1000-grain weight - increased yield for m^2^ (field trials)	[113]
	*OsCKX2* knockout obtained by CRISPR/Cas9	rice (*Oryza sativa*)	- increased shoot fresh and dry weight both in normal-phosphate and low-phosphate conditions - lesser leaf yellowing and increased maximum quantum efficiency of PSII under Pi deficiency - increased P concentration in roots and shoots under low-Pi conditions	[114]
	*proAGIP::GhCKX3b* silencing construct carpel- and stamen-specific promoter	cotton (*Gossypium hirsutum*)	- increased seed number - increased lint yield	[115]
glucosyl-transferase	*35S::ZOG1*	tobacco (*Nicotiana tabacum*)	- primary root elongation and diminished branching	[116]
	*Act1::cZOGT1* *Act1::cZOGT2*	rice (*Oryza sativa*)	- short shoots - delayed leaf senescence	[117]
	*Ubi1::ZOG1*	maize (*Zea mays*)	- delayed leaf senescence - shorter stature, thinner stems, narrower leaves - increased root biomass and branching - disturbed floral development, smaller ear	[118]
	*35S::UGT85A5*	tobacco (*Nicotiana tabacum*)	- increased tolerance to salt stress: increased seed germination rate on the medium containing NaCl (100–200 mM); increased total fresh weight of salt-treated seedlings (exposed to 100 or 200 mM NaCl for 4 weeks) - lesser decay of Chl content in leaf discs incubated in 100–300 mM NaCl - survival under strong salt-induced stress (seedlings watered with NaCl solution increasing to 300 mM for 4 weeks) - increased proline content and decreased MDA content in salt-treated plants (watered with 300 mM NaCl for 1 week)	[119]
	*35S::OscZOG1* *OsZOG1* silencing construct under *Ubi* promoter	rice (*Oryza sativa*)	- in overexpressing line: improved growth of lateral roots, decreased shoot growth and yield-associated traits, accelerated senescence - in silenced line: improved crown roots growth and tillering, higher shoots, increased yield-associated traits: panicle branching, grain number per panicle, seed size, and 1000-grain weight	[120]
	*35S::AtUGT76C2*	thale cress (*Arabidopsis thaliana*)	- reduced tolerance to osmotic stress at postgermination stage: decreased germination rate, slower germination and primary root growth, more severe stress symptoms (seeds sown on medium with 200, 250, 300 mM mannitol) - increased tolerance to drought at mature stage: less severe stress symptoms, improved survival rate (plants not watered for 7 days), decreased water loss and faster stomata closure in detached leaves	[121]
	*Ubi::AtUGT76C2*	rice (*Oryza sativa*)	- enhanced root growth - increased sensitivity to abiotic stresses during germination and post-germination growth: slower germination and decreased seedling growth (seeds exposed to 100 mM NaCl, 7.5% PEG8000, or 150 mM mannitol) - increased tolerance to salt stress: less severe stress symptoms, decreased electrolyte leakage, increased survival rate, increased proline and soluble sugar content (seedlings watered with 200 mM NaCl for 2 weeks), lower H_2_O_2_ and O_2_^•−^ level, increased activity of SOD, CAT, and APX (plants exposed to 200 mM NaCl for 12 h) - increased drought tolerance: less severe stress symptoms, increased survival rate, increased proline and soluble sugar content (seedlings not watered for 1 week), decreased water loss from detached leaves	[122]

## 5. Gibberellins

Gibberellins are a large group of tetracyclic diterpenoids. Among them, only a few compounds participate in the regulation of growth and development of higher plants—primarily GA_1_ and GA_4_ [2]. Gibberellin deficiency causes dwarfism [123]. For arable crops, especially cereals, dwarfism can be an advantage, because it improves lodging resistance and changes assimilate partitioning so that more assimilates are allocated to flowers and grains. Breeding of semi-dwarf cereal varieties has been proven to be enormously successful in increasing grain yield since the advent of the “green revolution” [124]. The rice *semidwarf-1* (*sd-1*) gene, encoding gibberellin 20 oxidase, is well known as the “green revolution gene” and is considered to be the one of the most important genes deployed in modern rice breeding. It has contributed to the significant increase in crop production that occurred in the 1960s and 1970s, especially in Asia [125,126]. The genes responsible for the “green revolution” in wheat are semi-dwarfing genes *Reduced height* (*Rht*). The most important and widely used are the alleles *Rht-B1b* and *Rht-D1b* that are found in >70% of current commercial wheat cultivars. They are known to reduce stem extension by causing partial insensitivity to gibberellins due to the changed signaling process [127,128].

The crucial enzymes associated with gibberellin metabolism are GA20ox and GA3ox gibberellin oxidases, catalyzing the last two steps of the synthesis of biologically active gibberellins, as well as GA2ox oxidase, catalyzing the oxidation of these phytohormones to inactive catabolites. The three types of enzymes mentioned above are encoded by small gene families, *GA20ox*, *GA3ox,* and *GA2ox,* respectively [129]. All of them were targets of genetic engineering. The introduction of *GA20ox* and *GA3ox* or *GA2ox* genes enables us to obtain plants with either increased or decreased active gibberellin content (Table 3). Increased gibberellin content stimulates elongation growth and lignin synthesis, while a reduced level of these phytohormones results in dwarfism, stimulation of lateral shoot formation, and reduction of lignin content (Table 3). Change in gibberellin content also allows us to obtain other useful traits. Tomato fruits with *GA20ox1* gene overexpression remained firm for a longer time, which prolonged their shelf life [130]. The formation of longer xylem fibers in transgenic poplars overexpressing *AtGA20ox1* is beneficial for paper production [131]. On the other hand, reduced lignin content in transgenic switchgrass with decreased gibberellin content facilitates the bioethanol production process. However, the *GA2ox*-overexpressing lines were semi-dwarf, which is not a desired trait in plants grown for biomass production [132]. Enhanced activity of gibberellin biosynthetic enzymes may be not beneficial in terms of resistance to certain pathogens. Transgenic rise overexpressing *OsGA20ox3* was more susceptible to *Xanthomonas oryzae* pv. *oryzae* (causing bacterial blight) and *Magnaporthe oryzae* (causing rice blast), while *OsGA20ox3* knockdown lines displayed enhanced resistance to these pathogens [133].

Modification of gibberellin signaling was also taken into consideration. DELLA protein was identified as a repressor in the gibberellin signaling pathway. The above-mentioned *Rht* alleles encode DELLA proteins [128]. Canola mutant *ds-3* bearing a mutation in the gene encoding DELLA protein is semi-dwarf [134]. The overexpression of *SLR1* encoding rice DELLA protein enhanced cold tolerance in this species, while plants with overexpression of *GA20ox1* were more sensitive to this kind of stress. These results suggest that weakening of gibberellin signaling leads to the improvement in chilling tolerance [135]. Overexpression of *GoGID1* encoding gibberellin receptor in alfalfa allowed to obtain transgenic plants with increased growth rates, heights, and biomass production when compared to the control [136].

**Table 3 plants-11-03430-t003:** Summary of the results of the experiments on transgenic plants with changed gibberellin concentration. CAT, catalase; Chl, chlorophyll; GAMT, gibberellin methyltransferase; GAox, gibberelin oxidase; MDA, malonyldialdehyde; POX, peroxidase; RWC, relative water content; SOD, superoxide dismutase; WUE, water-use efficiency.

Enzyme	Promoter and Gene	Species	Phenotype of Transgenic Plants Compared to Control Lines	Reference
**Modifications aimed at increasing of active gibberellins level**				
GA20ox	*35S::AtGA20ox1*	hybrid aspen (*Populus tremula* × *P. tremuloides*)	- increased growth rate - increased biomass - longer and more numerous xylem fibers	[131]
	*35S::AtGA20ox1*	tobacco (*Nicotiana tabacum*)	- shoot growth stimulation - increased biomass production - increased lignin content - stimulation of xylem formation	[137]
	*35S::CcGA20ox1*	tomato (*Lycopersicon esculentum*)	- changed morphology: higher shoots, non-serrated leaves, some flowers had longer style - delayed flowering - increased fruit number and their total weight per plant, some of the fruits were parthenocarpic, which was not observed in the control plants - delayed fruit ripening	[130]
	*Ubi1::AtGA20ox1*	maize (*Zea mays*)	- higher and more slender stems - increased vegetative biomass - increased content of lignin and cellulose	[138]
	*35S::PdGA20-OXIDASE* *DX15::PdGA20-* *OXIDASE* poplar xylem-specific promoter	hybrid poplar (*Populus tremula* L. × *Populus alba*)	- increased shoot growth and biomass production, accompanied by poor root growth and unstable shoots in plants with constitutive overexpression - enhanced growth without changing the overall phenotype in plants with xylem-specific expression	[139]
GA3ox	*35S::StGA3ox2* *StLS1::StGA3ox2* leaf-specific promoter *Tub1::StGA3ox2* tuber-specific promoter	potato (*Solanum tuberosum*)	- plants with constitutive overexpression or leaf-specific overexpression were higher and tuberized earlier when grown under short day conditions - plants with tuber-specific overexpression did not display differences in shoot height, their tuberization was slightly delayed - increased tuber biomass per plant in line with constitutive overexpression	[129]
**Modifications aimed at decreasing of active gibberellins level**				
GA2ox	*Act::OsGA2ox1* *D18::OsGA2ox1* promoter of a gene participating in gibberellin synthesis in rice	rice (*Oryza sativa*)	- plants with constitutive overexpression displayed changed morphology: dwarfism, darker green, broader and shorter leaves, they also failed to set grains - lines with expression under D18 promoter were semi-dwarf and developed normal flowers and grains	[140]
	*35S::AtGA2ox8*	canola (*Brassica napus*)	- dwarfism - increased anthocyanin content in leaves - stimulated branching, increased number of siliques - increased seed yield per experimental plot	[141]
	*35S::OsGA2ox5*	thale cress (*Arabidopsis thaliana*) rice (*Oryza sativa*)	- dwarfism, delayed onset of generative phase - increased starch granule accumulation and gravity responses - increased tolerance to salt stress: improved survival rate and less reduced seedling growth (seeds exposed to 100 mM or 140 mM NaCl for 7 days); significantly improved survival rate (seedlings exposed to 170 mM NaCl)	[142]
	*rbcs::PtGA2ox1* leaf-specific promoter *TobRB7::PtGA2ox1* root-specific promoter *LMX5::PtGA2ox1* stem-specific promoter	tobacco (*Nicotiana tabacum*)	- slower growth - increased drought tolerance: increased RWC, increased proline and sugar content, decreased MDA content, elevated POX, SOD, and CAT activities (plants not watered for 19 days)	[143]
	*Ubi1::PvGA2ox5* *Ubi1::PvGA2ox9*	switchgrass (*Panicum virgatum*)	- semi-dwarfism - increased tillering - decreased lignin content	[132]
	*Ubi::GA2ox6* mutated versions	rice (*Oryza sativa*)	- changed morphology: reduced plant height, expanded root system, enhanced tillering - increased WUE and photosynthesis rate - increased grain yield (field trials) - increased drought tolerance: less pronounced wilting symptoms; improved leaf recovery; increased survival rate; increased proline content and activities of SOD, CAT, POX; decreased H_2_O_2_ content (plants were air-dried for 6 h then recovered for 6 days) - increased tolerance to salt stress: increased survival rate (plants exposed to 200 mM NaCl for 2 days) - increased tolerance to temperature (both heat and cold) stress: increased survival rate (plants exposed to 4 °C or 42 °C for 2 days) - increased tolerance to biotic stress: less pronounced symptoms of infection with the bacterium *Xanthomonas oryzae* pv. *oryzae*, increased seedling weight after infection with the fungus *Pythium arrhenomanes*, limited spread of fungus *Fusarium fujikuroi*	[144]
	*35S::GhGA2ox1*	cotton (*Gossypium hirsutum*)	- increased drought tolerance: increased proline, Chl, and RWC in stressed plants (plants not watered for 10 days) - increased tolerance to salt stress: increased proline, Chl, and RWC in stressed plants (plants watered with 200 mM NaCl for 10 days)	[145]
GAMT	*35S::AtGAMT1*	tomato (*Solanum lycopersicum*)	- changed morphology: reduced plant height, smaller leaves of changed shape - smaller stomata - increased drought tolerance: delayed wilting symptoms, higher leaf water status, reduced transpiration, improved recovery of stressed plants (plants not watered for 14 days)	[146]
CYP cytochrome 450	*ProEui::PtCYP714A3* native promoter of one of *CYP* genes in rice	rice (*Oryza sativa*)	- semi-dwarfed phenotype, promoted tillering, reduced seed size - increased tolerance to salt stress: less severe stress symptoms, improved survival rate (seedlings exposed to 150 mM NaCl for 12 days)	[147]
	*35S::CYP71D8L*	rice (*Oryza sativa*)	- dwarfed phenotype, reduced grain number per panicle - increased drought tolerance: less severe stress symptoms, increased Chl content and decreased H_2_O_2_ level after 5 days of drought (plants not watered for 10 days) - increased tolerance to salt stress: less severe stress symptoms, increased Chl content and decreased H_2_O_2_ level after 3 days of stress (plants watered with 150 mM NaCl for 8 days)	[148]

## 6. Brassinosteroids

Similar to other phytohormones, brassinosteroids have a pleiotropic effect and they participate both in the regulation of development and stress response [149]. Considering their chemical structure, these compounds belong to polyhydroxy steroids and are similar to animal steroid hormones [150]. Many experiments concerning the impact of brassinosteroids on plant development and stress tolerance were carried out by spraying the plants with solutions of these phytohormones. There are also data in the literature on genetic engineering of the pathway of their biosynthesis [151]. Important determinants of yield that are regulated by brassinosteroids are plant height, leaf angle, and inflorescence architecture [152]. The results of the experiments on transgenic plants with increased brassinosteroid content are shown in Table 4. There were also experiments aiming at decreasing the level of these phytohormones. Overexpression of *AtBAT1* encoding brassinosteroid-inactivating acyltransferase in bentgrass resulted in decreased growth rate, erect leaves, prolonged leaf longevity, and improved drought resistance [153].

The rice mutant *ebisu dwarf* (*d2*) with impaired brassinosteroid biosynthesis had erect leaves, which improves light penetration into the canopy. On the other hand, it produced smaller seeds [154]. However, another rice mutant, *osdwarf4-1,* had a stature similar to the *d2* mutant, while the morphology of its flowers and seeds remained unchanged [155,156]. Barley accessions carrying a single recessive gene *uzu*, encoding the brassinosteroid receptor, are semi-dwarf. This gene has been introduced in almost all Japanese hull-less barley cultivars [157,158]. Brassinosteroid insensitive semi-dwarf barley mutants were more tolerant to drought [159].

Transgenic tomatoes overexpressing *SIBRI1* encoding brassinosteroid receptor displayed increased height, yield, and fruit number per plant. Transgene overexpression also caused an increase in the levels of carotenoids, ascorbic acid, soluble solids, and soluble sugars during fruit ripening [160]. On the other hand, it led to a decrease in drought tolerance [161]. Rice line overexpressing *SERK2*, encoding membrane protein interacting with brassinosteroid receptor, produced larger grains and was more tolerant to salt stress [162]. Overexpression of kinases participating in brassinosteroid-induced signaling, e.g., membrane localized ZmBSK1 and downstream acting ZmCCaMK, in corn, led to the improved drought tolerance [163]. Overexpression of the gene *TaBRI1* from wheat, encoding transmembrane receptor kinase, in Arabidopsis resulted in increased sensitivity to brassinosteroids, earlier flowering, and increased silique size and seed yield [164]. While the modification of signaling through BRI receptors is linked to growth arrest, the overexpression of BRL3, a vascular-enriched member of the brassinosteroid receptor family, in *A. thaliana* enhanced tolerance to drought without penalizing plant growth [165]. In recent years, there has been significant progress in the deciphering of brassinosteroid signaling, which opens the way to successful modification of crop plants [152].

**Table 4 plants-11-03430-t004:** Summary of the results of the experiments on transgenic plants with increased brassinosteroids concentration due to overexpression of hydroxylases participating in the biosynthesis of these phytohormones. CAT, catalase; MDA, malonyldialdehyde; POX, peroxidase; RWC, relative water content; SOD, superoxide dismutase.

Enzyme	Promoter and Gene	Species	Phenotype of Transgenic Plants Compared to Control Lines	Reference
C-22α hydroxylase	*35S::AtDWF4*	thale cress (*Arabidopsis thaliana*)	- stimulation of generative shoot branching - increase in siliques and seed number	[166]
	expression of cDNA of CYPs from maize, *A. thaliana*, and rice under AS promoter active in stems, leaves, and roots	rice (*Oryza sativa*)	- enhanced tillering - increased grain yield per plant, increased grain weight (greenhouse and field trials) - increased leaf angle, more loose stature (not desired trait)	[167]
	*35S::CYP724B1*	rice (*Oryza sativa*)	- increased spikelet number per panicle - increased grain size and 1000-grain weight - increased leaf angle, more loose stature	[168]
	*35S::AtDWF4*	canola (*Brassica napus*)	- longer roots, increased root biomass - larger leaves - stimulation of branching, increase in siliques number - increased seed yield per plant - increased drought tolerance: improved survival rate, increased root and shoot biomass of recovered plants (plants not watered for 12 days) - increased tolerance to heat stress: improved survival rate of stressed plants (plants exposed to 45 °C for 3 or 4 h) - increased resistance to necrotrophic fungal pathogens *Leptosphaeria maculans* and *Sclerotinia sclerotiorum:* less pronounced symptoms of infection	[169]
	*35S::SoCYP85A1*	tobacco (*Nicotiana tabacum*)	- longer primary root and more lateral roots - enhanced drought tolerance: less severe stress symptoms, higher RWC, decreased water loss rate, increased proline content in one transgenic line, decreased MDA content and H_2_O_2_ level, increased activity of SOD, CAT, and POX (plants not watered for 10 days)	[170]
	*35S::* *PtCYP85A3*	tomato (*Lycopersicon esculentum*) poplar (*Populus davidiana ×* *P. bolleana*)	- promoted growth and biomass production - increased plant height, shoot fresh weight and fruit yield in tomato - increased plant height and stem diameter, enhanced xylem formation in poplar	[171]
	*Ubi::OsDWF4* *Gt1::OsDWF4* seed-specific promoter	rice (*Oryza sativa*)	- enhanced tillering - increased grain yield per plant, slightly increased 1000-grain weight - increased leaf angle, more loose stature (not desired trait)	[172]
	*Ubi::ZmDWF4*	maize (*Zea mays*)	- bigger ears, improved grain yield per ear - increased 1000-grain weight - faster growth, increased plant height and node number - increased leaf area, delayed leaf senescence	[173]
	*35S::PeCPD*	poplar (*Populus tomentosa*)	- increased plant height, biomass, stem diameter and xylem formation - increased tolerance to salt stress: less visual injuries, lower H_2_O_2_ and O_2_^•−^ formation, decreased MDA content, increased levels of soluble proteins and proline, increased SOD activity (plants exposed to 50 mM NaCl for 3 days and then to 100 mM NaCl for 12 days)	[174]
cytochrome catalyzing conversion of 6-deoxocastasterone to castasterone	*35S::DWF*	tomato (*Lycopersicon esculentum*)	- faster germination and seedling growth - increased plant height and weight - slender stature, leaf deformations - faster ripening of fruits - decrease in fruit yield per plant but increase when normalized on the cultivated area due to higher density of plants	[175]
	*35S::DWF*	tomato (*Lycopersicon esculentum*)	- increased tolerance to chilling stress: lesser amount of oxidized proteins, lower level of lipid peroxidation and electrolyte leakage, increased maximum quantum efficiency of PSII, increased activity of APX and enzymes participating in ascorbate and glutathione recycling (plants exposed to 4 °C for 3 days)	[176]
enzyme catalyzing the conversion of 6-deoxocathasterone and 3-dehydroteasterone to 6-deoxotyphasterol and typhasterol, respectively	*Ubi::TaD11-2A*	rice (*Oryza sativa*)	- increased grain length and 1000-grain weight - increased starch content and decreased amylose content	[177]

## 7. Abscisic Acid

The most important functions of ABA include regulation of dormancy, stomata opening, as well as maturation and germination of seeds. This phytohormone also participates in the response to abiotic stress, primarily drought [2]. In higher plants, carotenoids, specifically violaxanthin or neoxanthin, are substrates for ABA biosynthesis. An important enzyme necessary for the synthesis of violaxanthin (and indirectly neoxanthin) is zeaxanthin epoxidase (ZEP). In the ABA biosynthetic pathway, both xanthophylls are converted to the conformation 9-*cis*, and then 9-*cis*-epoxycarotenoid dioxygenase (NCED) catalyzes xanthoxin formation. Later, xanthoxin undergoes two-step oxidation—first to abscisic aldehyde and then to ABA. These reactions are catalyzed by short-chain alcohol dehydrogenase/reductase (SDR) and abscisic aldehyde oxidase (AAO), respectively [178]. Seo et al. [179] observed that under stress conditions AAO expression does not change, while the expression of the *LOS5/ABA3* (*LOS5*) gene is enhanced. This gene encodes an enzyme responsible for sulphation of AAO molybdenum cofactor [180]. Therefore, experiments on plants overexpressing the *LOS5* gene were also conducted.

The results of the experiments on transgenic plants with increased ABA content are collected in Table 5. They indicate that the modifications associated with this phytohormone are a very promising direction of research aimed at obtaining varieties with increased drought tolerance. On the other hand, it was reported that ABA overproducing transgenic tomato was significantly more vulnerable to xylem embolism [181]. Apart from the synthesis and degradation of this phytohormone, ABA transport also has an impact on stress tolerance. The *Lr34res* gene conferring durable resistance to multiple fungal pathogens in rice was reported to be an ABA transporter [182].

ABA-induced signaling pathways are a subject of intensive research. The expression of tomato genes encoding ABA receptors belonging to PYR/PYL/RCAR family in *A. thaliana* improved drought tolerance of transgenic plants [183]. Similarly, overexpression of native genes of subfamily III of PYR/PYL/RCAR family in *A. thaliana* resulted in increased ABA-sensitivity and enhanced drought resistance [184]. Overexpression of *OsPYL3* and *OsPYL9* in rice enhanced tolerance to cold stress and drought; and similarly, overexpression of *TaPYL4* in wheat improved drought tolerance [185]. Transgenic poplars overexpressing *PtPYRL1* or *PtPYRL5* were more tolerant to drought, cold, and osmotic stress [186]. Membrane-bound kinase OsPKR15 was shown to interact with OsPYL11, an orthologue of AtPYL9. Ectopic expression of OsPKR15 in *A. thaliana* increased its sensitivity to ABA and resulted in the enhancement of drought tolerance [187]. The potential of ABA receptors overexpression for the improvement of water-use efficiency (WUE) in crops was proposed by Mega et al. [188]. Rice overexpressing *OsPYL6* under the control of *Arabidopsis thaliana* Responsive to Dehydration 29A *(AtRD29A*) promoter displayed enhanced tolerance to dehydration. On the other hand, the reduced grain yield under non-stress conditions due to reduction in height, biomass, panicle branching, and spikelet fertility was also observed in transgenic plants [189]. The role of SNF 1-RELATED PROTEIN KINASE 2 (SnRK2), comprising a subfamily of plant-specific protein kinases, in ABA signaling and stress tolerance is being investigated [190]. Overexpression of wheat genes of *TaSnRK2s* in *A. thaliana* resulted in improved tolerance to drought, salt, and cold stress [185]. Overexpression of *ARR5,* encoding one of the SnRK2 targets, in *A. thaliana* resulted in ABA hypersensitivity and enhanced drought tolerance [191]. Transgenic *A. thaliana* expressing *TaCIPK27*, encoding a wheat kinase involved in stress response, displayed enhanced ABA-sensitivity and improved drought tolerance [192].

**Table 5 plants-11-03430-t005:** Summary of the results of the experiments on transgenic plants with increased abscisic acid concentration. APX, ascorbate peroxidase; CAT, catalase; LOS, sulfurase of molybdenum cofactor required for abscisic aldehyde oxidase activity; MDA, malonyldialdehyde; NCED, 9-*cis*-epoxycarotenoid dioxygenase; POX, peroxidase; RWC, relative water content; SOD, superoxide dismutase; WUE, water-use efficiency; ZEP, zeaxanthin epoxidase.

Enzyme	Promoter and Gene	Species	Phenotype of Transgenic Plants Compared to Control Lines	Reference
ZEP	*35S::AtZEP*	thale cress (*Arabidopsis thaliana*)	- increased tolerance to salt stress: increased fresh weight of stressed plants (seedlings exposed to 0–160 mM NaCl for 10 days) - increased tolerance to osmotic stress: increased fresh weight of stressed plants (seedlings exposed to 0–400 mM mannitol for 10 days) - increased drought tolerance: drought survival (plants not watered for 3 weeks, control plants died); reduced water loss from detached shoots	[193]
	*35S::MsZEP*	tobacco (*Nicotiana tabacum*)	- increased tolerance to salt stress: increased content of soluble sugars and proline, increased activity of SOD, and decreased content of MDA (plants watered with 200 mM NaCl for 2 weeks) - increased drought tolerance: less pronounced wilting symptoms, increased content of soluble sugars and proline, increased activity of SOD and decreased content of MDA (plants not watered for 2 weeks)	[194]
	*EsABA1* under the control of artificial superpromoter	tobacco (*Nicotiana tabacum*)	- increased tolerance to salt stress: increased shoot dry weight and total root length in stressed plants (seedlings exposed to 250 mM NaCl for 4 weeks) - reduced Chl degradation in leaf discs incubated in 400 and 600 mM NaCl solutions for 3 days	[195]
NCED	*35S::AtNCED3*	thale cress (*Arabidopsis thaliana*)	- increased drought tolerance: lower transpiration rate, less pronounced wilting symptoms in stressed plants (plants not watered for 18 days)	[196]
	*35S::VuNCED1*	creeping bentgrass (*Agrostis stolonifera*)	- increased tolerance to salt stress: increased fresh and dry biomass and less pronounced wilting symptoms in stressed plants, increased survival rate (plants watered with 0.2–0.8% NaCl for 10 weeks) - increased drought tolerance: increased fresh and dry biomass, less pronounced wilting symptoms in plants exposed to water deficit, increased survival rate (reduced watering for 10 weeks)	[197]
	*35S::OsNCED3*	thale cress (*Arabidopsis thaliana*)	- delayed seed germination, slower growth, changed leaf morphology - sugar oversensitivity - increased drought tolerance: less severe stress symptoms (plants not watered for 9–13 days)	[198]
	*35S::CrNCED1*	tobacco (*Nicotiana nudicaulis*)	- increased tolerance to salt stress: reduced Chl degradation and H_2_O_2_ and O_2_^•−^ generation in leaf discs incubated in 200 mM NaCl solution for 4 days - increased drought tolerance: reduced water loss from detached leaves, higher turgor and increased RWC in stressed plants (plants not watered for 1 week), lower H_2_O_2_ and O_2_^•−^ generation in leaves subjected to 80 min dehydration - increased tolerance to oxidative stress: reduced Chl degradation in leaf discs incubated in 1% H_2_O_2_ solution for 4 days	[199]
	*35S::OsNCED4*	thale cress (*Arabidopsis thaliana*)	- delayed seed germination, slower growth, changed morphology - sugar oversensitivity - increased drought tolerance: less severe stress symptoms (plants not watered for 9–13 days)	[200]
	*35S::OsNCED3*	rice (*Oryza sativa*)	- promotion of leaf senescence (darkness induction protocol) - increased drought tolerance: increased survival rate of stressed plants (seedlings not watered for 18 days) - increased tolerance to salt stress: increased survival rate of stressed plants (seedlings exposed to 150 NaCl)	[201]
	*35S::VaNCED1*	grapevine (*Vitis vinifera*)	- changed leaf morphology - increased drought tolerance: less severe stress symptoms (plants not watered for 50 days)	[202]
	*PvNCED1* under dexamethasone-inducible promoter	tobacco (*Nicotiana plumbaginifolia*)	- increased drought tolerance: reduced water loss from detached leaves with induced transgene expression; less pronounced wilting symptoms in plants with induced transgene expression (plants not watered for 10 days)	[203]
	*HvLea::AtNCED6* drought-responsive promoter	barley (*Hordeum vulgare*)	- improved performance under water deficit: higher RWC and CO_2_ assimilation rate, improved WUE (plants maintained at 10% soil moisture level for 4 days, stress imposed after anthesis)	[204]
	*rd29A::LeNCED1* stress-responsive promoter	petunia (*Petunia hybrida*)	- increased drought tolerance: less pronounced wilting symptoms in stressed plants, reduced water loss, significantly increased survival rate (plants not watered for 2 weeks)	[205]
LOS	*AtLOS5* under the control of artificial superpromoter	tobacco (*Nicotiana tabacum*)	- increased drought tolerance: reduced water loss from detached leaves; less pronounced wilting symptoms, increased activity of CAT and APX and increased proline content in stressed plants (plants not watered for 6 days)	[206]
	*AtLOS5* under the control of artificial superpromoter	cotton (*Gossypium hirsutum*)	- increased drought tolerance: reduced water loss from detached shoots; less pronounced wilting symptoms in stressed plants (plants not watered for 5 days) - increased fresh weight; SOD, POX, and APX activities; and proline content; decreased MDA content in plants exposed to reduced watering for 8 weeks	[207]
	*AtLOS5* under the control of artificial superpromoter	maize (*Zea mays*)	- increased drought tolerance: reduced water loss from stressed plants and increased survival rate (plants not watered for 2 weeks) - increased SOD, CAT, and POX activities and proline content, decreased content of H_2_O_2_ and MDA in plants exposed to reduced watering for 5 days	[208]
Hydroxylase participating in ABA catabolism	RNAi-mediated suppression of the *Hv8′ hydroxylase* construct expressed under drought-responsive promoter	barley (*Hordeum vulgare*)	- improved performance under water deficit: higher RWC and CO_2_ assimilation rate, improved WUE (plants maintained at 10% soil moisture level for 4 days, stress imposed after anthesis)	[204]
	RNAi-mediated suppression of the *OsABA8ox1*	rice (*Oryza sativa*)	- increased tolerance to alkalinity: increased survival rates, decreased membrane injury, MDA, H_2_O_2_, and O_2_^•−^ content in roots (seedlings were exposed to 10, 15, 20 mM Na_2_CO_3_) - increased survival rate, less severe stress symptoms, more vigorous growth, increased Chl content, increased panicle number, spikelets per panicle, percentage of filled spikelets and 1000-grain weight (seedlings were transplanted into soil of pH 7.59, 8.86, and 9.29) - increased grain yield per plant under salt stress (plants grown in soil of pH = 9.29, EC = 834.4 µS cm^−1^)	[209]

Many of the ABA-responsive transcription factors have been identified to date, among them, those belonging to NAC, bZIP, AP2/ERF, and WRKY families [210]. Stress-responsive grapevine transcription factor VvNAC17 was shown to increase sensitivity to ABA and drought tolerance when its gene was overexpressed in Arabidopsis [211]. Similarly, the overexpression of soybean *GmNAC019* in *A. thaliana* led to the hypersensitivity to ABA and higher survival rate in a soil-drying assay [212]. The overexpression of drought-induced maize *ZmWRKY26* in *A. thaliana* improved its tolerance to drought and heat [213]. The positive role of *MaWRKY80* from banana in drought stress resistance was shown in the experiment with transgenic Arabidopsis. Among other effects, this transcription factor modulated the expression of genes encoding ABA biosynthetic enzymes [214]. *Capsicum annuum* ABA Induced ERF (*CaAIEF1*) expressed in *A. thaliana* enhanced drought tolerance of transgenic plants [215]. The overexpression of *VlbZIP30*, encoding a transcription factor belonging to the bZIP family in grapevine, in transgenic *A. thaliana* improved dehydration tolerance [216]. It was shown that corn transcription factor ZmbZIP33 interacts with core components of ABA signaling. Its overexpression in Arabidopsis led to the increase in ABA content and drought tolerance [217]. Arabidopsis plants overexpressing *TabZIP14-B* from wheat exhibited enhanced tolerance to salt and cold, as well as increased ABA sensitivity [218]. A maize gene *ZmMYB3R*, encoding MYB transcription factor, is known to be induced by ABA. Its overexpression in *A. thaliana* caused increased sensitivity to ABA and enhanced tolerance to drought and salt stress [219]. Increased sensitivity to ABA resulting in the enhanced tolerance to drought, salt, and osmotic stress was also observed in *A. thaliana* with overexpression of another transcription factor from maize, *ZmHDZIV14* [220]. Dehydration responsive element binding factors (DREB) belong to the AP2/ERF family. The expression of ABA-induced *AhDREB1* from peanuts in *A. thaliana* resulted in increased ABA levels and increased sensitivity to this phytohormone, as well as in improved drought tolerance [221]. ZmPTF1 transcription factor, belonging to the bHLH family, is known to be a positive regulator of ABA synthesis. Its overexpression in maize caused an increase in ABA content and enhanced drought tolerance [222]. The other examples of genetic modification of ABA receptors, ABA signaling components, and ABA-responsible transcription factors can be found in the reviews [223,224,225].

## 8. Ethylene

Ethylene is another phytohormone important for the regulation of the stress response. Among its other functions, the one important for farmers is the stimulation of fruit ripening [226]. Due to the simple structure of the molecule and ethylene occurrence in the gas phase, this compound is often applied exogenously. Treatment with the ethylene precursor, 1-amino-3-cyclopropane-1-carboxylic acid (ACC), was also applied [227].

As ethylene is known to be a plant growth inhibitor, many of the experiments aimed to decrease the synthesis of this phytohormone (Table 6). The main target of genetic engineering is 1-amino-3-cyclopropane-1-carboxylic acid synthase (ACS), responsible for the synthesis of the direct precursor of this phytohormone. This enzyme was discovered to be crucial for the regulation of ethylene biosynthesis. Partial silencing of the expression of ACS encoding genes in maize resulted in higher yields of transgenic lines compared to control when plants were exposed to drought [228]. Apple and melon fruits with decreased ACS activity ripened more slowly and were firmer than the fruits of non-transformed plants, which is a desirable trait if there is a need for longer storage [229,230]. Interestingly, inoculation of pea with the soil bacterium *Variovorax paradoxus* synthesizing ACC deaminase (ACC decomposing enzyme) resulted in improved growth and seed yield under drought conditions when compared to plants inoculated with the *V. paradoxus* mutant, in which ACC deaminase activity was significantly lower [231]. 

Inactivation of *ZmACO2* encoding ACC oxidase2 catalyzing the final step of ethylene biosynthesis via genome editing using CRISPR/Cas9 method led to the reduction of ethylene production in developing ears and increased grain yield per ear [239].

There were also attempts to modify ethylene-induced signaling. The ethylene response factor superfamily is known to participate in response to various environmental stresses, such as drought, salt, heat, and cold. An elegant summary of the research on ERFs, their participation in stress response, and their genetic engineering using CRISPR/Cas9 genome editing tool was written by Debbarma et al. [240]. The modification of ERF-dependent signaling turned out to be promising also in the case of improving biotic stress response. Transgenic rice overexpressing *OsERF83* was more resistant to *Magnaporthe orizae*, causing one of the most destructive diseases in rice [241]. Overexpression of *GmERF3* in tobacco resulted in increased tolerance to drought and salt stress but also enhanced resistance to the bacteria *Ralstonia solanacearum*, fungus *Alternaria alternata*, and tobacco mosaic virus [242]. Transgenic *A. thaliana* with overexpression of *MbERF12* from *Malus baccata* displayed enhanced antioxidant response and increased tolerance to low temperature and salt stress [243]. Transgenic lines of *A. thaliana* overexpressing *ERF1* were more tolerant to drought, salt, and heat stress [244]. It was shown that ectopic constitutive expression of *ERF95* and *ERF97* led to the increase in tolerance to the heat stress in *A. thaliana* [245]. Overexpression of native ERF in rubber tree resulted in the stimulation of root growth, increased dry biomass, and increased tolerance to salt stress [246]. Overexpression of *TdSHN1,* encoding cDNA of SHINE-type ERF transcription factor from durum wheat, in tobacco improved tolerance to Cd, Cu, and Zn [247]. Overexpression of *MdERF1B* from the apple tree significantly enhanced cold tolerance of *Arabidopsis thaliana* seedlings, and transgenic apple seedlings and calli [243]. Transgenic tomato overexpressing *SlERF5* was more tolerant to drought and salt stress [248]. Other examples of modulation of ethylene signaling resulting in the enhancement of the tolerance to salt stress were reviewed by Riyazuddin et al. [249]. Members of the ARGOS family are known to be negative regulators of ethylene responses. Genetic engineering targeted at ARGOS8, including both overexpression and modification by the CRISPR/Cas 9 method, was used to obtain maize with improved grain yield under drought stress conditions [250]. The regulatory role of miRNA in stress response and its connection with ethylene signaling were also elucidated. For example, salinity-induced miR319 was reported to positively regulate ethylene synthesis and increase tolerance to salt stress in switchgrass [251].

## 9. Jasmonic Acid and Its Derivatives

Jasmonates play a role in plant response to various stress factors, including biotic ones [252]. They also participate in the regulation of plant development. The exogenous application of methyl jasmonate increased the yield of soybean [253]. Application of JA or JA together with gibberellin GA_3_ resulted in an increase in ginseng yield [254]. The administration of JA alleviated the adverse effects of salt stress on rice and barley [255,256]. However, there is inconsistency in the results reported in the literature, as improved salt tolerance was also observed in transgenic plants with enhanced JA degradation [257]. JA is also used as an elicitor in the production of various secondary metabolites [258].

The experiments on transgenic plants with changed content of JA or its methyl ester were also carried out (Table 7). Among them, interesting ones concern plants overexpressing jasmonic acid carboxyl methyltransferase (JMT), which converts JA into its methyl ester. Another strategy is obtaining plants with overexpression of 13-lipoxygenase or 12-oxophytodienoate reductase participating in the biosynthesis of this phytohormone. Such modifications resulted in an increased tolerance to selected biotic and abiotic stresses and stimulation of the growth of underground storage organs.

Jasmonic acid signaling is an object of intensive research [274]. For example, overexpression of TdTIFY11a, a member of TIFY protein family participating in JA signaling, from *Triticum durum* in *A. thaliana* promoted germination under salt stress [275]. Expression of *VaNAC17*, encoding *Vitis amurensis* transcription factor, known to be induced by drought stress, in *A. thaliana* resulted in enhanced JA synthesis and drought tolerance [276]. Similarly, the overexpression of *VaNAC26* improved tolerance to drought and salt stress in *A. thaliana* [277]. The overexpression of *OsbHLH034* gene encoding transcription factor acting as positive regulator in JA signaling resulted in the increased resistance to rice bacterial blight, but it also increased sensitivity to salt stress [278]. The overexpression of *OsbHLH148* improved drought tolerance in transgenic rice [279]. Heterologous overexpression of JA-responsive transcription factor from *Ipomea batatas IbMYB116* in *A. thaliana* caused upregulation of the expression of JA biosynthetic genes, promoted JA accumulation and the JA response, and improved the tolerance to drought stress [280]. Enhanced proline accumulation and increased drought tolerance were also observed in soybean overexpressing another JA-responsive transcription factor, *GmTGA15* [281]. The modulation of the expression of JAZ proteins that are negative regulators of JA signaling allow us to obtain plants more tolerant to salinity and drought [257,282].

## 10. Future Perspectives

The examples presented here indicate that modification of phytohormone metabolism and signaling is a promising direction of research aimed at the improvement of crop productivity and stress tolerance. The progress in this field is possible due to broadening of the knowledge concerning the regulation of plant growth, development, and stress response, as well as due to the improvement of the methodology used. Many genes that can be targets of genetic engineering have been identified up to date [7]. The extensive research aiming at deciphering phytohormone signaling pathways is being carried out. The modification of this signaling at various levels, from elements of signaling cascades, through transcription factors to miRNAs, is a very promising direction of genetic engineering of crop plants.

Considering the methods of genetic engineering, the most promising innovation is genome editing using the so-called CRISPR/Cas9 system [283]. The system is based on nucleases that can be relatively easily programmed to search for specific DNA sequences. Available variants of effector nucleases allow various modifications of the target region. This makes CRISPR/Cas9 a fast, effective, and precise genome editing tool [283]. It is used both to discover functions of certain genes and to obtain plants of potential application in agriculture. CRISPR/Cas9 genome editing seems especially promising in research aimed at modulation of cytokinin levels [284].

Intensive research on the regulation of gene expression led to the discovery of many promoters specific to certain tissues, organs, or stage of plant development. The application of these promoters allows better control of the time and site of transgene expression. Scientists also designed artificial promoters [285]. There are systems enabling us to combine and introduce multiple genes at once (such as the Golden Gate modular cloning box), as well as methods for the introduction of large DNA fragments into plant cells. New successful protocols of crop species transformation are being developed [7].

An important obstacle in obtaining transgenic plants with improved yield is the well-known trade-off between stress defense and plant growth. One of its reasons is the energetic cost of the development and maintenance of various protective mechanisms, both biochemical and morphological. However, the negative effect of defense induction on growth often results from antagonistic crosstalk between phytohormones rather than from an identified metabolic expenditure. Sometimes, it is caused by pleiotropic effects of certain resistance traits or is a consequence of genetic linkage [286]. Therefore, it is possible to reduce the costs of plant defense. The strategies aimed at such a reduction were summarized by Karasov et al. [286].

To date, the majority of studies on transgenic lines with altered phytohormone content or signaling have been conducted under laboratory conditions. To obtain improved varieties suitable for regular cultivation, it is necessary to carry out large-scale field tests to determine whether the modifications introduced allow us to obtain the desired phenotype under natural conditions. At the same time, care should be taken to minimize the risk of transgene leak, so that genetically modified varieties would not pose the threat of contamination to the genomes of closely related wild species.

## Figures and Tables

**Table 6 plants-11-03430-t006:** Summary of the results of the experiments on mutant and transgenic plants with changed ethylene concentration. ACC, 1-aminocyclopropane-1-carboxylic acid; ACS, 1-aminocyclopropane-1-carboxylic acid synthase; CAT, catalase; ERF1, ethylene response factor 1; POX, peroxidase; RWC, relative water content; SOD, superoxide dismutase.

Protein	Promoter and Gene	Species	Phenotype of Transgenic Plants Compared to Control Lines	Reference
ACS	mutant *acs7*	thale cress (*Arabidopsis thaliana*)	- slightly faster germination - faster growth at the vegetative stage - increased tolerance to salt stress: improved survival of salt-treated seedlings (exposed to 150 mM NaCl); germination in presence of 150 mM NaCl slowed down to a lesser extent - increased tolerance to osmotic stress: germination in presence of 300 mM mannitol slowed down to a lesser extent - increased tolerance to heat stress: lower percentage of chlorosis in stressed seedlings (exposed to 43 °C for 3 h)	[232]
	*35S::ACS* antisense target silencing	apple tree (*Malus pumila*)	- firmer fruits - increased shelf-life	[229]
	*35S::PmACS* antisense target silencing	melon (*Cucumis melo*)	- firmer fruits - slower ripening	[230]
	*ZmUbi1::ZM-ACS6* hairpin target silencing	maize (*Zea mays*)	- increased grain yield of some lines in locations where drought occurred (field trials) - increased yield under low-nitrogen treatment (field trials)	[228]
	*ZmUbi1::ScACS1* *ZmUbi1::ScACS2* *ZmUbi1::ScACS3* hairpin target silencing	sugarcane hybrid cultivar (*Saccharum officinarum* × *Saccharum spontaneum*)	- increased plant height, leaf length, and leaf area - reduced carbon assimilation - no reduction in Chl content or sucrose levels - induction of non-enzymatic antioxidant apparatus	[233]
ACC deaminase	*35S::ACCD* from bacteria Other promoters used were: root-specific promoter of *rolD* gene from *Agrobacterium rhizogenes*, pathogenesis-related promoter of *prb-1b* gene from tobacco	tomato (*Lycopersicon esculentum*)	- increased tolerance to flooding, especially in lines where root-specific promoter was used: increased shoot fresh and dry weight, increased Chl content (plants flooded for 9 days)	[234]
	*35S::ACCD* from *Pseudomonas putida,* Other promoters used were: root-specific promoter of *rolD* gene from *Agrobacterium rhizogenes*, pathogenesis-related promoter of *prb-1b* gene from tobacco	canola (*Brassica napus*)	- increased tolerance to salt stress in lines where root-specific promoter was used: increased shoot and root dry weight, increased protein and Chl content (seedlings treated with 0–200 mM NaCl for 6 weeks of growth)	[235]
	*35S::TaACCD* from fungus *Trichoderma asperellum*	thale cress (*Arabidopsis thaliana*)	- improved root growth: increase in root length and root number, increase in total fresh weight and RWC (but no difference in dry weight), increase in seed number per pod - improved tolerance to salt stress: less severe stress symptoms, increased root length, number, and weight, increased RWC, decreased H_2_O_2_ content, later occurrence of O_2_^•−^ level increase and cell damage, decreased electrolyte leakage, less pronounced decrease in Chl content, increased POX activity (seedlings watered with 150 mM NaCl for 8 days)	[236]
	*35S::acdS* gene of *Pseudomonas veronii*	thale cress (*Arabidopsis thaliana*)	- slightly improved tolerance to salt stress: improved seedling growth (seeds exposed to 0, 50, 100 mM NaCl, seedlings grown for 25 days) - improved tolerance to flooding: less severe stress symptoms (plants water-logged for 5 days)	[237]
	*35S::ACCD* from *Achromobacter xylosoxidans*	geranium (*Pelargonium graveolens*)	- increased tolerance to salt stress: less pronounced decrease in Chl content, higher CO_2_ assimilation rate, lower content of H_2_O_2,_ increased activity of SOD, CAT, and POX (plants exposed to 50–200 mM NaCl for 30 days) - increased drought tolerance: less pronounced decrease in Chl content, higher CO_2_ assimilation rate, lower content of H_2_O_2_, increased activity of SOD, CAT and POX (reduced watering or ceasing watering for 15 days)	[238]

**Table 7 plants-11-03430-t007:** Summary of the results of the experiments on transgenic plants with increased concentration of jasmonic acid or its methyl ester. AOC, allene oxide cyclase; JMT, jasmonic acid carboxyl methyltransferase; LOX, lipooxygenase; MDA, malonyldialdehyde; OPR, 12-oxophytodienoate reductase; SOD, superoxide dismutase.

Enzyme	Promoter and Gene	Species	Phenotype of Transgenic Plants Compared to Control Lines	Reference
**Enhancement of jasmonic acid biosynthesis**				
LOX	*35S::* *TomloxD*	tomato (*Lycopersicon esculentum*)	- increased tolerance to heat stress: less pronounced wilting symptoms and faster recovery (seedlings exposed to 50 °C for 2 h) - increased resistance to fungal infection: less pronounced symptoms of infection by *Cladosporium fulvum*	[259]
	*35S::* *TomloxD*	tomato (*Lycopersicon esculentum*)	- decreased herbivore insect feeding (plants exposed to *Helicoverpa armigera*) - increased resistance to necrotrophic pathogen: less severe visual symptoms of infection with *Botrytis cinerea*	[260]
	*35S::CmLOX10*	thale cress (*Arabidopsis thaliana*)	- increased drought tolerance: less severe wilting symptoms, increased survival rate, lower electrolyte leakage, H_2_O_2_ and MDA level (plants not watered for 10 d) - decreased stomatal aperture and water loss from leaves	[261]
	*35S::TgLOX4* *35S::TgLOX5*	thale cress (*Arabidopsis thaliana*)	- longer leaves in *TgLOX5* overexpressing line, wider leaves - increased plant height - stimulated branching	[262]
AOC	*35S::TaAOC1* *Ubi:TaAOC1*	thale cress (*Arabidopsis thaliana*) wheat (*Triticum aestivum*)	- shorter roots - increased activity of SOD - improved tolerance to salt stress: less pronounced reduction in root growth of transgenic wheat (seedlings were treated with increasing concentrations of NaCl for 4 days then exposed to 200 mM NaCl for the next 4 days); increased survival rate of transgenic Arabidopsis (plants were treated with increasing concentrations of NaCl for 4 days, then exposed to 200 mM NaCl for the next 2 weeks) - improved tolerance to osmotic stress and oxidative stress in transgenic Arabidopsis: no or very small reduction in root length (seedlings exposed to 100, 200, 300 mM mannitol or 1, 2 mM H_2_O_2_ for 10 d)	[263]
	*35S::TaAOS*	tobacco (*Nicotiana benthamiana*)	- enhanced tolerance to Zn: lesser decrease in Chl content in leaf discs exposed to 10 and 20 mM ZnCl_2_ for 6 days	[264]
	*Ubi::AhAOC*	rice (*Oryza sativa*)	- increased plant height and root length - improved tolerance to salt stress: less pronounced reduction in seedling root growth (seeds germinated in presence of 80 or 120 mM NaCl); less severe stress symptoms, less pronounced reduction in plant height, increased content of proline and soluble sugars (plants exposed to 120 mM NaCl for 2 weeks)	[265]
	*35S::GhAOC1*	thale cress (*Arabidopsis thaliana*)	- enhanced tolerance to Cu: higher survival rate, increased shoot fresh weight and photosynthetic efficiency, reduced cell membrane damage and lipid peroxidation (plants watered with 120 μM CuCl_2_ for 10 days)	[266]
OPR	*35S::ZmOPR1*	thale cress (*Arabidopsis thaliana*)	- increased tolerance to salt and osmotic stress: improved germination in the presence of NaCl or mannitol (seeds exposed to 100–200 mM NaCl or 100–500 mM mannitol), more seedlings remained green (observations made after 7 days) - no differences in survival rate when older seedlings were exposed to NaCl (300 mM NaCl for 10 days)	[267]
	*Ubi1::AtOPR3*	wheat (*Triticum* *aestivum*)	- changed timing of development: delayed germination, slower growth, late flowering, delayed senescence - increased tolerance to short-term freezing: higher maximum quantum efficiency of PSII and decreased electrolyte leakage (plants were transferred to 4 °C for 24 h; their detached leaves were then subjected to decreasing temperatures to final 1 °C, −2 °C, or −5 °C and then incubated for 24 h)	[268]
**Enhancement of methyl jasmonate biosynthesis**				
JMT	*35S::* *AtJMT*	thale cress (*Arabidopsis thaliana*)	- increased resistance to fungal infection: lack of severe infection symptoms in plants 3 days after spraying with *Botrytis cinerea* spores	[269]
	*35S::* *AtJMT*	thale cress (*Arabidopsis thaliana*)	- increased resistance to infection with bacteria *Pseudomonas syringae* and enhanced expression of defense genes	[270]
	*35S::BcNTR1*	soybean (*Glycine max*)	- increased drought tolerance: less pronounced wilting symptoms, plant survival (plants not watered for 6 days, all control plants died); reduced water loss from detached leaves - increased tolerance to osmotic stress: increased fresh biomass of seedlings germinating in the presence of 0.3 M mannitol	[271]
	*35S::* *AtJMT*	potato (*Solanum tuberosum*)	- stimulated tuberization, increased tuber size - increased tuber yield per plant	[272]
	*35S::* *AtJMT*	ginseng (*Panax ginseng*)	- root growth stimulation - increased content of protopanaxadiol group of ginsenosides	[273]

## Data Availability

Not applicable.

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
