# Peer review of "Modifications of Phytohormone Metabolism Aimed at Stimulation of Plant Growth, Improving Their Productivity and Tolerance to Abiotic and Biotic Stress Factors"

_plants, 2022, doi:10.3390/plants11243430_

Round 1
Reviewer 1 Report
This manuscript is a comprehensive summary of the genes involved in each plant hormone and their usefulness to agriculture across a wide range of plant species.I have several comments below.
Cytokinin section.
The authors should mention the molecular species of cytokinins. In particular, the fact that tZ and iP have different effects and that tZ is synthesized in the root and transported through the vascular bundle to the shoots would be important information for CK-based breeding.
In the table of cytokinin
An antisense fragment DNA of the 3'-untranslated region of OsCKX2 is used in rice CKX2 (Ashikari et al., 2005), not cording sequence.
GA section
Although not modified by genetic engineering, I think it would be useful for the authors to mention the success of SD1 in rice and Rht in wheat to introduce the reader to the significant impact of GA on agriculture. Similarly, I would recommend introducing the BR receptor uzu for semi-dwarfism in barley (rather than just introducing improved drought tolerance).
Author Response
Respected Reviewer,
Thank you very much for all the remarks and suggestions for the manuscript improvement.
I have introduced corrections and extended the paper according to the review. I have also read the whole text once more and corrected some minor things (like spelling mistakes). Changed fragments are marked with the red font.
Thank you for the valuable suggestions concerning adding information that would make the review more informative.
Cytokinin section: The authors should mention the molecular species of cytokinins. In particular, the fact that tZ and iP have different effects and that tZ is synthesized in the root and transported through the vascular bundle to the shoots would be important information for CK-based breeding.
Thank you for this suggestion, the information about cytokinin types has been added.
In the table of cytokinin: In antisense fragment DNA of the 3'-untranslated region of OsCKX2 is used in rice CKX2 (Ashikari et al., 2005), not cording sequence.
Thank you for noticing, I was misled by the fragment “however, transgenic plants with antisense strands of OsCKX2 that had reduced levels of expression developed higher grain numbers”. I have changed this.
GA section: Although not modified by genetic engineering, I think it would be useful for the authors to mention the success of SD1 in rice and Rht in wheat to introduce the reader to the significant impact of GA on agriculture. Similarly, I would recommend introducing the BR receptor uzu for semi-dwarfism in barley (rather than just introducing improved drought tolerance).
Thank you for suggesting it, indeed this information was needed to fully show and emphasize the impact of phytohormones in crop improvement. I have added the information about rice, wheat in barley in gibberellins and brassinosteroids subchapters.
I hope that the improvements introduced would make the review acceptable for publication.
Yours faithfully,
Beatrycze Nowicka
Reviewer 2 Report
1. In the abstract add few concluding sentences that highlights the importance of this study.
2. Remove the word ‘publication’ at line 14 and replace with more appropriate word ‘review or study’
3. Add a section about “how genetic/metabolic modification was carried out? Define strategies briefly that better justify the title of manuscript.
4. In column 4 of Table 2 unify the style, explain in bullets or separate outcomes with comma (Adopt anyone style).
5. Moreover, the methodology is not required in all tables, only results are enough.
Author Response
Respected Reviewer,
Thank you very much for all the remarks and suggestions for the manuscript improvement.
I have introduced corrections and extended the paper according to the review. I have also read the whole text once more and corrected some minor things (like spelling mistakes). Changed fragments are marked with the red font.
- In the abstract add few concluding sentences that highlights the importance of this study.
The concluding sentence was added.
- Remove the word ‘publication’ at line 14 and replace with more appropriate word ‘review or study.
The word was replaced.
- Add a section about “how genetic/metabolic modification was carried out? Define strategies briefly that better justify the title of manuscript.
The suggested section was added to the manuscript (Subchapter 2).
- In column 4 of Table 2 unify the style, explain in bullets or separate outcomes with comma (Adopt anyone style).
This was improved. Sometimes the multiple effect were mentioned after common bullet point, separated with commas, when they were referring to the increased/decreased response to the certain stress type.
- Moreover, the methodology is not required in all tables, only results are enough.
It’s true that these parts were making the tables unnecessarily longer. However, I thought that some information about the experiments may be valuable (especially about the longevity of the stress). Therefore, I shortened the information concerning the methodology.
I hope that the improvements introduced would make the review acceptable for publication.
Yours sincerely,
Beatrycze Nowicka